Time series data analysis and ARIMA modeling to forecast the short-term trajectory of the acceleration of fatalities in Brazil caused by the corona virus (COVID-19)

James Akini akini.james1@my.uwi.edu
Tripathi Vrijesh
Department of Mathematics and Statistics, Faculty of Science and Technology, University of the West Indies St. Augustine , Port of Spain , Trinidad and Tobago
Shang Yilun
Electronic publication date: 2021 Jul 15
Publication date: 2021
Volume: 9
Electronic Location ID: e11748
Received 2020 Oct 5; Accepted 2021 Jun 19
Copyright: © 2021 James and Tripathi
Copyright year: 2021
Copyright holder: James and Tripathi
License: This is an open access article distributed under the terms of the Creative Commons Attribution License, which permits unrestricted use, distribution, reproduction and adaptation in any medium and for any purpose provided that it is properly attributed. For attribution, the original author(s), title, publication source (PeerJ) and either DOI or URL of the article must be cited.
License URL: https://creativecommons.org/licenses/by/4.0/

Keywords: Velocity, Acceleration, Fatalities, Infections, Composite function, Chain rule, Derivative, Trajectory

Funding: The authors received no funding for this work.

==============================
Objective

This paper incorporates the concept of acceleration to fatalities caused by the coronavirus in Brazil from time series data beginning on 17th March 2020 (the day of the first death) to 3rd February 2021 to explain the trajectory of the fatalities for the next six months using confirmed infections as the explanatory variable.

Methods

Acceleration of the cases of confirmed infection and fatalities were calculated by using the concept of derivatives. Acceleration of fatality function was then determined from multivariate linear function and calculus chain rule for composite function with confirmed infections as an explanatory variable. Different ARIMA models were fitted for each acceleration of fatality function: the de-seasonalized Auto ARIMA Model, the adjusted lag model, and the auto ARIMA model with seasonality. The ARIMA models were validated. The most realistic models were selected for each function for forecasting. Finally, the short run six-month forecast was conducted on the trajectory of the acceleration of fatalities for all the selected best ARIMA models.

Results

It was found that the best ARIMA model for the acceleration functions were the seasonalized models. All functions suggest a general decrease in fatalities and the pace at which this change occurs will eventually slow down over the next six months.

Conclusion

The decreasing fatalities over the next six-month period takes into consideration the direct impact of the confirmed infections. There is an early increase in acceleration for the forecast period, which suggests an increase in daily fatalities. The acceleration eventually reduces over the six-month period which shows that fatalities will eventually decrease. This gives health officials an idea on how the fatalities will be affected in the future as the trajectory of confirmed COVID-19 infections change.

Introduction

Background

The coronavirus (COVID-19) outbreak caused by a virus known as severe acute respiratory syndrome coronavirus 2 (SARS-CoV-2), originated in Wuhan, China and as of 30th June 2020, there have been 10.6 million confirmed cases, 5.8 million recovered cases, and 513 thousand deaths recorded worldwide. On 30th June 2020, Brazil recorded the 2nd highest number of confirmed cases of the coronavirus in the world and the 2nd highest number of fatalities in the world. This record shows that Brazil had 1.4 million confirmed cases, 790 thousand recovered cases and 59 thousand deaths from the coronavirus (World Health Organization, 2020).

During the pandemic, various countries have implemented stringent strategies in their fight against the virus. Since the novel coronavirus started to spread across the world, governments have issued lockdowns, stay-at-home orders, and curfews to help contain the outbreak. However, in Brazil, there was a more relaxed approach from the country’s president which in turn has affected the spread of the virus (McLaughlin, 2020).

Statement of the problem

To assess the possible future of the pandemic, in particular the fatalities, it is helpful to look not just at the number of cases, but also at how quickly they are increasing (Cohn et al., 2020). Most research focuses on modelling the changes in the death toll due to the coronavirus by observing its behavior based on socioeconomic and demographic factors and not necessarily on the impact of the confirmed infections.

Objective of study

This paper incorporates the concept of acceleration on the fatalities caused by the coronavirus in Brazil from time series data beginning from 17th March 2020 (the day of the first death) to 3rd February 2021 to explain the trajectory of fatalities for the next six months using the confirmed infections as the explanatory variable, that is, the impact of confirmed infections on the number of deaths. Thereafter, forecasting is done to observe the behavior of deaths based on the number of confirmed infections over the next six months.

Velocity and acceleration

Velocity is the rate of change of fatalities which is the first order derivative of the number of deaths per day (Utsunomiya et al., 2020; Chen & Yu, 2020). However, measuring the speed of the fatalities provides little information about the acceleration in the fatalities. The acceleration functions are much more sensitive and can be better utilized to provide useful information in real time to monitor, evaluate and forecast the COVID-19 epidemic in Brazil (Chen & Yu, 2020). Acceleration is the slope of the velocity as a function of time, which is the second derivative. It explains how rapid the change in velocity of fatalities is (Muncaster, 1993). Multivariate linear models as well as the calculus chain rule for composite functions were used to explain the acceleration of fatalities with respect to acceleration of confirmed infections. The chain rule gives us a way to calculate the derivative of a composite function. The chain rule principles state that if fatalities and confirmed infections are two functions then to get the derivative (velocity) of the composite function (death as a function of confirmed cases) simply divide the derivative of fatalities by the derivative of confirmed infections. Similarly, this theory is applied to derive the acceleration function by dividing velocity functions (Nykamp, 2020).

Auto regressive integrated moving average (ARIMA)

The Auto Regressive Integrated Moving Average (ARIMA) time series model is a reliable method frequently used in data analysis to forecast times series data and as a result, the acceleration of fatality functions can be used to produce ARIMA models for forecasting purposes. ARIMA models are used for non-stationary data and is made up of the Auto Regression model, AR(p), which uses the dependent relationship between Yt and p number of lagged observations included in the model, also called the lag order. The model also includes the Integrated (I) aspect, which is the differencing of raw observations, d times to allow for the time series to become stationary. Making the time series data stationary is necessary since stationary series are relatively easier to predict. Most statistical forecasting is based on the fact that the times series can be approximately stationarized through the use of mathematical transformations. The predictions for the stationarized series can then be “untransformed” by reversing whatever mathematical transformations used, to obtain predictions for the original series (Nau, 2020). If a time series dataset has a stable long-run trend and tends to revert to the trend line following a disturbance, it may be possible to stationarize it by de-trending (removing the effects of trends). However, in such a dataset like the corona virus in Brazil, sometimes even de-trending is not sufficient to make the series stationary, in which case it may be necessary to transform it into a series of period-to-period and/or season-to-season differences. That is, differencing the time series. Stationarizing a time series through differencing (where necessary) is an important part of the process of fitting an ARIMA model. The difference order, d, is the differences between consecutive observations. The last aspect of the ARIMA model includes the Moving Average model MA(q) which uses the dependency between Y1t and a residual error from a moving average model applied to q lagged observations. Hence, the time series model is denoted as ARIMA (p,d,q) model (Chen, 2019; Stock & Watson, 2007; Box et al., 2016).

Purpose of the study

The acceleration of fatality ARIMA models were used for forecasting the pandemic to come to a reasonable understanding of the trajectory of deaths in the short run. Short-term forecasts generated can be useful to guide the allocation of resources that are critical to bring the epidemic under control (Roosa et al., 2020).

Review of literature

Throughout the period of the pandemic, numerous studies have been conducted to better analyze and make predictions regarding the end of the pandemic. Researchers have developed models to make forecast about the fatalities due to the coronavirus.

Researchers have used the count of daily cases to formulate ARIMA models to predict the end of the pandemic (Bayyurt & Bayyurt, 2020; Dehesh, Mardani-Fard & Dehesh, 2020). One study adopted three kinds of mathematical models, that is, logistic model, Bertalanffy model and Gompertz model where the logistic model was the best fit among the three models. In this research, the epidemic trends of SARS were first fitted and analyzed to prove the validity of the existing mathematical models. The results were then used to fit and analyze the situation of COVID-19. This is the case since COVID-19 and SARS virus are both variants of coronaviruses and the infection pattern may be similar (Jia et al., 2020). The logistic model was used to explore the risk factors and predict the probability of occurrence of the disease according to the risk factors. It can predict the development and transmission law of epidemiology.

Many other study designs have made predictions based on compartmental models, with the population divided into classes and with assumptions being made about the rate of transfer from one class to another. These models are mostly differential equation prediction models. They employed mathematical modeling techniques to study the transmission and spread of COVID-19 to predict the magnitude and timing of the epidemic peak and the final epidemic size under various intervention strategies. Carcione et al. (2020) used the Susceptible-Exposed-Infectious-Removed (SEIR) model to describe the spread of the virus and compute the number of infected and dead individuals. Their model aimed to compute the number of infected, recovered, and dead individuals based on the number of contacts, probability of disease transmission, incubation period, recovery rate and fatality rate. Okhuese (2020) used a similar approach to propose a mathematical model for the end in the spread and subsequent elimination of the virus by using a new deterministic endemic model (Susceptible-Exposed-Infectious-Removed–Undetectable-Susceptible: SEIRUS). The study combined quarantine observatory procedures and behavioral change social distancing in the control and eradication of the disease in the most exposed sub-populations.

Veloso & Ziviani (2020) conducted a study by modelling the country level death toll velocity and acceleration. They used factors such as the daily COVID-19 death toll curve in each country, country’s countermeasures in response to the COVID-19 pandemic, community mobility reports, estimations of critical care beds available for and needed by COVID-19 patients in each country, and country’s development indicators.

Another researcher conducted a study on the velocity and acceleration, utilizing the concept of motion to observe the confirmed cases in China to explain the spread of the epidemic. This was then compared to the spread after massive interventions that took place in China (Chen & Yu, 2020).

Most studies used socioeconomic and demographic factors to explain the pandemic and thereafter, make projections based on these factors. However, this study seeks an alternative approach, by observing the confirmed infections since it is highly correlated to the fatalities. It is assumed that if the number of the confirmed infections can determine the nature of the fatalities, then better predictions can be made about the pandemic. This study borrows the concept of motion used by Chen & Yu (2020) and applies it to data from Brazil. This study does not make comparisons with the interventions that took place in Brazil to come to conclusions like most of the previously mentioned studies. The derivative functions represent the velocity and acceleration of fatalities based on the velocity and acceleration of confirmed infections, respectively. Our study has used the confirmed infections as the explanatory variable for fatalities. In addition, the concept of time series ARIMA modeling has been used to produce forecasts.

Materials & methods

Data

The data repository for the 2019 Novel Coronavirus was retrieved online from the Center for Systems Science and Engineering (CSSE) at Johns Hopkins University (Dong, Du & Gardener, 2020; Johns Hopkins University, 2020). The dataset contains data on the cumulative confirmed infections, cumulative recovered cases and the cumulative fatalities in Brazil starting from the 17th of March 2020, the day of the first recorded death. This is 21 days after the first recorded confirmed infection in Brazil. On the 17th of March 2020, Brazil already had a total of 321 confirmed infections and two recovered cases. The cumulative confirmed infections and cumulative fatalities show an upward trend. The dataset contained the COVID-19 cases for a period of approximately 11 months (46 weeks) in Brazil until the 3rd of February 2021 which is 344 days from the first confirmed infection and 324 days from the first death in Brazil. At the end of the study period, Brazil had already recorded 9,339,420 confirmed infections, 8,311,881 recovered cases and 227,563 fatalities. However, there are 799,976 active cases. Therefore, at this point there exist 89% of the patients recovered, 2.4% deaths and the remaining 8.6% still active cases.

Velocity and acceleration of the confirmed infections and fatalities

Time series data on the cumulative confirmed infections and cumulative fatalities were observed where each xi and zi are the daily confirmed infections and daily fatalities, respectively, for i = (21, 22, … t). i = 21 is the 21st day since the first recorded confirmed infection which is the day of the first recorded fatality in Brazil. Equation (1) below shows the general equation for the cumulative confirmed infections and cumulative fatalities.

(1) Y(y)=∫i=21t⁡yi=∑i=1t⁡yi

where Y(y) = {F(x) confirmed infections, H(z) confirmed fatalities}

However, F(x) and H(z) being extremely insensitive to changes in the pandemic, the first derivatives of Eq. (1) produces the general function, Y′(y), in Eq. (2) (Chen & Yu, 2020). These functions are said to be velocity functions which explains the change in the confirmed infections and change in fatalities caused by the pandemic in Brazil. That is, the new cases each day. This explains the speed of the confirmed infections and fatalities for i = (21, 22, … t). i = 21 is the 21st day since the first recorded confirmed infection which is the day of the first recorded fatality in Brazil.

(2) Y′(y)=∫i=21t+1⁡yi−∫i=21t⁡yi=∑i=21t+1⁡yi−∑i=21t⁡yi

where Y′(y) = {F′(x) velocity of confirmed infections, H′(z) velocity of fatalities}

Although F′(x) and H′(z) can measure the speed of the pandemic, they do not give any information regarding the acceleration, which is much more sensitive than the velocity functions (Chen & Yu, 2020). Thus, the second derivatives, F′′(x) and H′′(z) produced the general Eq. (3) below (Utsunomiya et al., 2020). This represents the acceleration of the confirmed infections and acceleration of fatalities each day in Brazil due to the pandemic, respectively (Chen & Yu, 2020). The acceleration explains how rapid is the change in velocity. If the acceleration functions produce zero, this is an early indication of neither acceleration nor deceleration of the pandemic. Where the acceleration functions produce values greater than zero, it is an early indication of acceleration of the pandemic and producing values less than zero indicates deceleration.

(3) Y′′(x)=Y′(xi+1)−Y′(xi)

where Y″(y) = {F″(x) acceleration of confirmed infections, H″(z) acceleration of fatalities}

Multivariate linear fatality function

Correlation analysis using ggpairs in R Programming software were used to produce the best fit multivariate linear functions from the data produced by the functions (Schloerke et al., 2020). Log transformation was done to meet the linear functions assumptions for normality (Feng et al., 2014). These functions were used to explain and forecast the velocity and the acceleration of the fatalities due to the coronavirus.

Velocity of fatality composite functions

Due to the obvious relation the confirmed infections have on the number of fatalities, the velocity of fatality composite functions of confirmed infections was utilized to formulate the rate at which deaths were occurring each day with respect to new confirmed infections named dH(z)dF(x) as shown in Eq. (4). This term explains the speed in the deaths influenced by the confirmed infections. The term begins at the date of the first recorded death, i = 21 which is the 21st day since the first recorded confirmed infection.

(4) dH(z)dF(x)=H′(z)÷F′(x)→dH(z)dF(x)=H′(z)×[F′(x)]−1

Acceleration of fatalities composite functions

The derivatives of the above velocity of fatality composite function formed the acceleration of fatality composite function as seen in Equation (5). That is, the term dH′(z)dF′(x), explains the rate at which the deaths accelerate with respect to the acceleration of the confirmed infections. This rate was used to explain and forecast the acceleration of fatalities caused by the pandemic. The term starts at the date of the first recorded death, i = 21 which is the 21st day since the first recorded confirmed infection.

(5) dH′(z)dF′(x)=H″(z)÷F″(x)→dH(z)dF(x)=H″(z)×[F″(x)]−1

Time series analysis

The data for acceleration of fatality functions were plotted using months as the horizontal axis. Data cleaning is often the first step that data scientists and analysts take to ensure statistical modelling is supported by good data (Hyndman, 2014). Therefore, the behavior of the acceleration of the fatalities within each month were then observed by plotting a monthly breakout of the data for fatality functions. This was done to observe the range of possible outliers within each month. Tsclean function in R programming software was used on the time series data for each of the acceleration of fatality functions to identify and replace any outliers or blanks with estimated values from the time series data using series smoothing and decomposition (Hyndman et al., 2020; Dalinina, 2017). Outliers are residuals that lie outside the range ±2(q0.9 – q0.1) where qp is the p quantile of the residuals. The residuals are identified by fitting a loess curve for non-seasonal data and via a periodic STL decomposition for seasonal data (Hyndman, 2014). However, this cleaned time series data, now referred to as the cleaned function, was graphed and observed. Only where there were extreme variances and volatility with the clean data, monthly (every 30 days from start) and weekly (every 7 days from start) moving averages (MA) were formed and compared to the cleaned data. Moving average is used to analyze data points by creating a series of averages of different subsets of the full data set to mitigate the impact of random short-term fluctuations over a specified timeframe (Hayes, 2020). That is to smooth out any noise or possible random outliers still present and to emphasize long term trends (Ross, 2019). However, given that the acceleration functions have cyclical patterns at most (bouncing upward and down) moving averages are not likely to capture meaningful trends thus over-smoothing the data (Smith, 2020). Hence, with the moving averages being too smooth, the cleaned data was used to form ARIMA models to forecast the trajectory of the velocity and acceleration of fatalities. The graphs produced from the two acceleration of fatality functions were plotted using months as the unit on the horizontal axis. Thereafter, the time series data was decomposed by splitting it into three components: seasonality, trends, and remainder. These intuitive components capture the historical patterns in the series and deconstructing a series into these components can help understand its behavior and prepare a foundation for building a forecasting model (Dalinina, 2017). Seasonality refers to patterns that repeat with a fixed period, trends are the underlying trend of the metrics and the remainder also known as the noise is the original time series after the seasonal and trend series are removed (Anomaly, 2020). This was done using the stl function in the R programming software (Cleveland et al., 1993). Seasonality over time, the trend line and the remainder of the models selected for forecasting in the previous step were observed. ARIMA models can be fitted to both seasonal and non-seasonal data. However, seasonal ARIMA requires a more complicated specification of the model structure. For this cause, we first attempted to de-seasonalize the series and use a non-seasonal ARIMA model (Ghosh, 2018; Dalinina, 2017). The function seasadj in the R programming software accomplishes this where it returns the seasonally adjusted data constructed by removing the seasonality (Hyndman et al., 2020).

The Augment Dikey Fuller Test (ADF) was then conducted on the selected cleaned data to test for stationarity using the adf.test function in the R programming software (Trapletti & Hornik, 2019). The ADF null hypothesis is that the variable has unit root (non-stationary) and the alternative hypothesis is that the variable does not have a unit root (Holmes, Scheuerell & Ward, 2020). If the p-value of the ADF test is small, it concludes stationarity of the data. On the other hand, where the printed p-values were large, we fail to reject the null hypothesis which means that the data is non-stationary (Hua, 2016). Stationarity and seasonality of the dataset can be further analyzed by using autocorrelation function (ACF) and partial autocorrelation function (PACF) graphs (Taspinar, Celebi & Tutkun, 2012). The cleaned data are plotted by using the ACF and PACF functions in R Programming (Hyndman et al., 2020). This plot shows the correlation between the time series and its lagged value. When the lines are outside the bounds of the ACF and PACF graphs, this suggests serious lags. According to how drastic the lag is, this will suggest non-stationarity. However, with the series being non-stationary, the data will be differenced until stabilization starting with a difference order of one and re-evaluation on whether further differencing is needed. This therefore eliminates (or reduces) trend and seasonality and converts the non-stationary series to a stationary series. Fitting an ARIMA model requires the series to be stationary since modeling a stable series with consistent properties involves less uncertainty. Hence, stationarity is required because only if the time series data is a deterministic (non-random) pattern, the research can use the ARIMA time series model (Hua, 2016). The ADF test is then conducted again but on the differenced data to test for its stationarity. ACF and PACF were plotted on the differenced data and the lags were observed.

Fitting the ARIMA model

The auto fit ARIMA model was produced on the de-seasonalized (without seasonality) cleaned data. This function uses a variation of the Hyndman-Khandakar algorithm which combines unit root tests, minimization of the Akaike Information Criterion (AIC) and Maximum Likelihood Estimation (MLE) to obtain an ARIMA model (Hyndman et al., 2020; Hyndman & Khandakar, 2008). The auto fit ARIMA model can produce a forecast, however, it must be checked to see if the model order parameter and structure are correctly specified ensuring that there are no significant autocorrelation present. This was done using the ACF and PACF plots, observing lags and adjusting the p or q to ensure that there are no significant autocorrelations present. That is, ensuring all lines were within the bounds of the PCF and PACF graphs. Additionally, models were also selected based on the Akaike Information Criterion (AIC). The AIC value for each model was observed and comparisons were made. This is a widely used measure of a statistical model. It basically quantifies the goodness of fit and the simplicity/ parsimony, of the model into a single statistic. When comparing models, the one with the lowest AIC is generally “better” (Keshvani, 2013; Kourentzes, 2016). The AIC was observed and compared amongst the auto ARIMA model and the adjusted model.

The adjusted model was validated for each of the functions using the holdout method by dividing the time series data into a training set and a testing set. The training set is what the model is trained on and the testing set is used to evaluate how well the model performs on unseen data (Kapil, 2018). A subset of the dataset (20%) was omitted from the ARIMA model. That is, a subset of 65 was set out of the dataset starting from case 260 to 324 to be used as the test set to see how well the model performs. It provides a final estimate of the machine learning model’s performance after it has been trained and validated. The remaining 80% of the data (no hold out) were then plotted. This is the training set, and it is what the model will be trained on. The selected ARIMA models for both acceleration functions were then forecasted for the rest of the study period using the forecast function in the R programming software to observe how the 95% and 80% forecasting interval fits the actual data points (Hyndman et al., 2020). The lines of the time series of the de-seasonalized counts were also plotted to compare the forecast with the actuals. Thereafter, we observe how the 95% and 80% forecasting interval fits the actual data points. If the validation shows that the model is not a good one, seasonality on the de-seasonalized cleaned data will be re-introduced and an ARIMA model would be produced on this. A model is not good if the line does not fall within the forecast interval and if the expected forecasts on the de-seasonalized model is too linear too soon, which is unlikely given the past behavior of the series (Dalinina, 2017). The AIC for each model was observed and comparisons were made. Based on this validation, the better ARIMA models for each function were used for forecasting the trajectory of the acceleration of fatalities for the next six months. A 30-day forecast was also done on the ARIMA models for each validated acceleration functions, mainly for comparison and observational purposes amongst the respective models. The ARIMA models for each validated function were the auto ARIMA model without seasonality, the adjusted lag model, and the Auto ARIMA model with seasonality. The most realistic ARIMA model for each validated function was then selected for forecasting the next six-month period.

Results

Acceleration of confirmed infections and fatalities

Figures 1A and 1B show the acceleration functions F′′(x) and H′′(z) derived in Eq. (3) and are the second derivative of F(x)and H(z), respectively using the ggplot function in the R programming software (Wickham et al., 2019). The acceleration of confirmed infections, F′′(x), shows continuous fluctuations in acceleration and deceleration where the magnitude of these changes increased until approximately the 119th day since the first confirmed infection (23rd June 2020). Thereafter the curve maintained its magnitude until 195th day since the first confirmed infection (7th September 2020) until 279th day (30th November 2020). Within such period there were random distinct spikes. The curve then gradually increased in magnitude throughout the rest of the study period, Fig. 1A.

Figure 1 Acceleration.

(A) Confirmed infections. (B) Fatalities

The acceleration of fatalities, H′′(z), have fluctuations where the horizontal axis represents the number of days from the first fatality, Fig. 1B. H′′(z), shows fluctuations between the acceleration and deceleration of fatalities about the zero line that gradually increased in magnitude until the 98th day since the first confirmed infection (2nd of June 2020) where this magnitude was then maintained until the 203rd day since the first confirmed infection (15th September 2020). Thereafter, the curve decreased in magnitude with few distinct spikes and then gradually increased throughout the rest of the study period.

Multivariate linear functions

The distributions and correlations on the raw data produced from the acceleration function in Eq. (3) were explored. This was done to understand the relationship amongst functions to conduct multivariate linear regressions for the acceleration of fatalities. However, producing linear functions requires the data for all functions to meet the normality assumptions previously mentioned. As a result, log transformation was done on the functions formed from Eq. (3) before producing linear functions. Due to the negatives produced in the acceleration function from Eq. (3), a common technique was used by adding a constant value to the data before applying the log transformation (Wicklin, 2011). The transformation for the acceleration of confirmed infections and fatalities was, therefore, log(F′′(x)+a1) and log(H′′(x)+a2), respectively where a1 and a2 are constants. These constants were chosen so that the min(F′′(x)+a1), and min(H′′(x)+a3) will be a small positive number, 0.001. As a result, the data was then transformed to log(F′′(x)+51285.001) and log(H′′(x)+974.001). Thus, the distribution and correlation on the transformed data was observed and normality of the functions were checked. The correlation between functions were used to form the multivariate linear function listed below.

The following acceleration of fatality linear function were produced using the acceleration of confirmed infections as the explanatory variables (Gardener, 2019):

Death_Acceleration1(F′′(x))=0.630929log(F′′(x)+51,285.001)

(6) R=0.9956

Death_Acceleration1 function has a coefficient of determination R2 = 99.56%. That is, 99.56% of the independent variable explains the acceleration of fatality suggesting that it is a good function.

The acceleration of fatality linear function designed in Eq. (6) was plotted over the study period which is approximately 46 weeks from the 9th to 54th week Fig. 2. The first week is when the first recorded confirmed infection took place whereas the 9th week is when the first recorded death occurred. Figure 2 shows that the Death_Acceleration1 function initially has a constant acceleration at zero until 1st April 2020, but the curve’s magnitude gradually increased until 3rd June 2020 where it maintained this average magnitude until 15th September 2020. The acceleration started reducing its magnitude thereafter until 24th November 2020 and then started largening throughout the rest of the study period.

Figure 2 Multivariate linear functions for Death_Acceleration1.

Acceleration of fatality composite functions

Figure 3 shows dH′(z)dF′(x), the acceleration of deaths as confirmed infections accelerate, and this function generally fluctuates a little above the zero line with three distinct pulses on 27th March 2020, 26th August 2020 and 22nd October 2020 that shows drastic deceleration of the fatalities and then the curve rapidly accelerated, returning to its average nature. The other noticeable pulses on 5th May 2020 and 5th June 2020 show sharp and large acceleration at first with an immediate deceleration back to its average nature.

Figure 3 Fatality composite functions for the acceleration in fatality with respect to the acceleration of confirmed infections.

Explanatory data analysis

Breakout of the data

A monthly breakout on the time series data produced from the above acceleration functions was done to observe the range of any possible outliers in the data points. Each of the functions of acceleration showed that there were very few noticeable outliers throughout the study period.

Cleaned functions and moving averages

However, to compensate for any possible outliers, the acceleration functions were first cleaned to produce cleaned functions for each. In addition to such, the weekly moving average (every 7 days from the start of the study period) and the monthly moving average (every 30 days from the start of the study period) functions were formed from the cleaned functions. The weekly and monthly moving averages were compared to their respective cleaned functions, and it showed that the moving averages were over-smoothing the data. That is, not capturing meaningful trends. Hence the cleaned functions were used for creating a model for forecasting.

Decomposition of the data

The cleaned function for each fatality function of acceleration was decomposed by extracting its seasonality, its trend line, and its remainder. Figure 4 shows the decomposition of the acceleration of fatality functions, along with its associated cleaned data where the horizontal axis represents the number of weeks since the first death in Brazil. Seasonality component was extracted from the cleaned data for each of the acceleration of fatality functions called the de-seasonalized data and this de-seasonalized series was used to create ARIMA models.

Figure 4 Decomposition of the cleaned data acceleration of fatality.

(A) Cleaned data for Death_Acceleration1; (B) cleaned data for composite function with respect to the change confirmed infections.

Stationarity

An ADF Test was firstly done on the cleaned data. This test outputs the Dickey Fuller value, Lag Order, and p-values. The smaller the Dickey Fuller value, the better the model (Holmes, Scheuerell & Ward, 2020). The Lag Order allows for higher-order autoregressive processes and the p-values conclude whether the model has unit root or not. The cleaned data for all the functions have small p-values which suggests stationarity of the functions. However, the ACF and PACF functions were used to verify this and lags for each function were observed.

Autocorrelation and choosing a model

ACF plots of the de-seasonalized series showed significant autocorrelations with many lags. The PACF plots showed that this could be due to carry-over correlation from the first lag in most cases. Hence, the de-seasonalized series was differenced until stabilization starting at difference order one. The ADF test was conducted again, on the difference data, and for all functions, it rejects the null hypothesis of non-stationarity. Table 1 shows the ADF test results for each acceleration of fatality functions and their difference order required to obtain stationarity and reduced lags. Both functions now show smaller Dickey Fuller values than the ADF test first conducted on the cleaned data and small p-values. Hence the differenced data is better.

Table 1 Augmented dickey fuller (ADF) test for differenced acceleration fatality functions.

	Difference order	Dickey fuller	Lag order	p-Value	
Death_Acceleration1	1	−14.488	6	0.01	
dH′(z)dF′(x)	1	−9.6114	6	0.01	

Thereafter ACF and PACF were produced for the differenced data for the acceleration functions to observe any spikes at specific lag points of the differenced series. It was found that the lags are not as drastic for the differenced data. This therefore suggests that the differencing of order one for the acceleration of fatality functions is sufficient and should be included in the model for each function.

Fitting the ARIMA model

Different models were fitted for each of the acceleration of fatality functions. The models are the Auto ARIMA Model without seasonality, a custom made ARIMA model called the adjusted lag model and the Auto ARIMA model with seasonality. The adjusted lag model compensates for the serious lags found, if any, in the auto ARIMA model without seasonality. Table 2 shows the fitted ARIMA models for the acceleration functions with their respective AIC values. The smaller the AIC value the better the model.

Table 2 Fitted ARIMA models for acceleration fatality functions.

Acceleration functions	Fit 1 (deseasonalized Auto.arima)	Fit 2 (Adjusted lag model)	Fit 3 (seasonalized Auto.arima)	
Death_Acceleration1	ARIMA(4,1,5)
AIC(−483.27)	ARIMA(7,1,5)
AIC (−556.66)	ARIMA(3,1,5)(0,0,2)[15]
AIC(−490.44)	
dH′(z)dF′(x)	ARIMA(1,1,4)
AIC (−1087.61)	ARIMA (7,1,4)
AIC(−1089.19)	ARIMA(1,1,3)(2,0,0)[15] with drift
AIC(−1092.49)	

Cross validation: holdout method

The Adjusted lag models were validated using the holdout method showing that the actual data were all within the 95% and 80% forecast interval which shows that it is a good model Fig. 5. However, the blue lines representing forecasts for the adjusted model for dH′(z)dF′(x) function seems unrealistic since the blue lines representing forecast in this function seems as though it might become linear too soon. That is the case since plotted predictions assume that there will be no other seasonal fluctuations in the data and the change in number of fatalities from one day to another is constant in mean and variance. Hence this forecast may be a naive model. As a result, seasonality was added back to this series and its predictions were also observed using the holdout method.

Figure 5 Cross validation using holdout method for acceleration of fatalities.

(A) Death_Acceleration1. (B) Composite function with respect to the change in confirmed infections.

The horizontal axes in Fig. 5 represents time in days. The vertical axes for the acceleration of fatality graphs represents the acceleration of fatality rates per acceleration of confirmed infections. The interpretation of the vertical axes gives little information due to the many manipulations of the data. As a result, we focus on the trajectory of the forecasts.

Further testing, selecting the best model, forecasting and analyzing

A further testing was done by observation, comparing a 30-day forecast on the ARIMA models listed in Table 2 for each of the acceleration functions; Death_Acceleration1 and dH′(z)dF′(x). ARIMA models were selected based on how realistic the model looked over the 30-day forecast. The better ARIMA model selected for the acceleration functions were the auto ARIMA model with seasonality. This is confirmed for dH′(z)dF′(x) where the auto ARIMA model with seasonality has the lowest AIC value. However, the decision made for the Death_Acceleration1 function goes against the AIC value which would have suggested that the adjusted lag model is better for making forecasts. Based on the 30-day forecast for such a model, the predicted curve was too constant, and its forecast interval started expanding like a funnel which seems a bit unrealistic when compared to the auto ARIMA model with seasonality. The summary on selecting the best ARIMA model for each function is summarized Fig. 6.

Figure 6 Acceleration of fatalities validated function for forecasting sequence diagram.

The six months forecasted ARIMA models for the multivariate linear function for the multivariate acceleration of fatalities function shows an increase and then a constant trajectory until where the magnitude of the acceleration gradually decreases over the forecasted period Fig. 7A. However, the composite function for the acceleration of fatalities shows large fluctuations at first but then gradually decreases in magnitude until it seems to eventually flatten (Fig. 7B).

Figure 7 Seasonalized auto ARIMA model six months’ forecast of acceleration of fatalities.

(A) Death_Acceleration1 (B) Composite function with respect to the change in confirmed infections.

Discussion

This study incorporates the concept of acceleration of fatalities caused by the coronavirus in Brazil to explain the projected trajectory of fatality for the next six months using the confirmed infections as the explanatory variable.

Two different functions were used to explain the acceleration of fatalities using the acceleration of the confirmed infections as the explanatory variable. One of the accelerations of fatality function is a multivariate linear function, Death_Acceleration1 and the other function,dH′(z)dF′(x), was formed in Eq. (5) by applying calculus chain rule for composite functions to produce the acceleration of fatality composite function.

Acceleration of cases, multivariate functions and composite functions

The acceleration in the confirmed infections generally increased in magnitude throughout the period. The acceleration eventually maintained an average range of values Fig. 1A. Closer observation showed that when the magnitude of the acceleration of the fatalities decreased over the period, the speed in the fatalities (velocity) slowed down. Hence, how rapid these fatality cases are occurring daily affects the outcome of the number of cases. Moreover, a close relationship can be seen between the confirmed infections and the fatalities since there exist similarities in their curves throughout the study period.

When the acceleration got smaller in magnitude, the change in fatalities started to reduce. On the other hand, where the speed in the change of the fatalities (acceleration) increased in magnitude, the change in the fatalities (velocity) started to increase. Although these functions consider the confirmed infections to explain the outcome of the fatalities, by observation, it shows to be a good representation of the actual fatalities’ functions produced from the raw data.

The multivariate function shows that the magnitude in acceleration increased over the period but eventually maintained in magnitude (Fig. 2). The composite function shows that the average size of the acceleration of the fatalities was basically small and constant throughout the period (Fig. 3). These functions developed in Figs. 2 and 3 seem to mirror the general results of the actual cases represented in Fig. 1B.

ARIMA modelling of functions and forecasting

The acceleration of fatality functions had very few distinct outliers throughout the study period. Nevertheless, the time series data for each function was still cleaned from any possible outliers and missing data points. Moving averages were expected to be taken on each of the cleaned functions to produce ARIMA models for forecasting. This would have worked well for our model since in reality new confirmed infections or changes in new confirmed infections at one point in time do not have an immediate impact on fatality and change in fatality, respectively. The moving averages provides for historical information. However, the moving averages for the acceleration of fatality functions are far smoother than its cleaned data. Using these moving averages can result in erroneous forecasts due to the data being too smooth which can cause a loss in information. Hence the cleaned functions produced were used instead for the ARIMA modelling to make forecasts.

The ARIMA models produced for each of acceleration of fatality functions are shown in Table 2. After conducting a cross validation (the holdout method) on the adjusted lag models and further testing of the ARIMA models by forecasting for the next 30 days, both acceleration models were found to be realistic to forecast for the next six months Fig. 5. The forecast in the acceleration composite function initially shows a large fluctuation in the acceleration function which suggests a quickening in daily fatalities. Thus, an increase in the velocity of fatalities in the earlier part of the six-month forecasted period is expected. However, the acceleration started to decrease which means that the rate at which fatalities were changing would eventually start to reduce. This model was replicated for São Paulo, one of the 26 states of the Federative Republic of Brazil. This state was chosen since it was the first state to record a Covid19 death in Brazil (Albuquerque, 2020). The forecast shows some similarities to the forecast made on the country Brazil where there exists some fluctuation in the beginning of the period until eventually flattening throughout the rest of the study period (Fig. 8).

Figure 8 Six months’ forecast of acceleration of fatalities for São Paulo.

(A) Multivariate function. (B) Composite function.

Limitations

This study had several limitations. There exists little referencing regarding the conclusions of this research since most journals did not forecast the accelerations of fatalities in Brazil by using confirmed infections as an explanatory variable. Also, there is little interpretation to the values in the y-axis for the forecast of the acceleration curves due to the many manipulations of the data. As a result, only the trajectory of the forecast was observed over time.

Conclusions

In general, the increased acceleration of fatalities in the earlier part of the forecasted period suggests an increase in daily fatalities. The increasing numbers in fatalities in Brazil during February 2021 and March 2021 confirms this prediction of increased fatalities as suggested by the acceleration model in this paper (Worldometer, 2021). It is expected by the model that the reduction in acceleration of fatalities will cause the new fatalities to slow down thus causing an eventual decrease in the fatalities over the next six-month period.

Supplemental Information

Supplemental Information 1 Cumulative confirmed infections, cumulative recovered cases and the cumulative fatalities in Brazil.

Click here for additional data file.

Supplemental Information 2 Code using R programming Software.

Click here for additional data file.

Additional Information and Declarations

Competing Interests

Author Contributions

Data Availability

The authors declare that they have no competing interests.

Akini James conceived and designed the experiments, performed the experiments, analyzed the data, prepared figures and/or tables, authored or reviewed drafts of the paper, and approved the final draft.

Vrijesh Tripathi conceived and designed the experiments, authored or reviewed drafts of the paper, and approved the final draft.

The following information was supplied regarding data availability:

The code is available in a Supplemental File.

The data are available at GitHub: https://github.com/CSSEGISandData/COVID-19/tree/d2b385d02de1fe8cb595fc33b243c065b97c2ddc/csse_covid_19_data/csse_covid_19_daily_reports.

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
