# Peer review of "Time series data analysis and ARIMA modeling to forecast the short-term trajectory of the acceleration of fatalities in Brazil caused by the corona virus (COVID-19)"

_PeerJ, doi:10.7717/peerj.11748_

## Round 0.1 · original submission · Major Revisions

The reviewers have provided detailed comments. Please address each point of these comments and prepare a response letter. Thank you.

Reviewer 1 ·

Basic reporting

This paper is well written. It borrows the concept of velocity and acceleration to forecast the coronavirus by using time series and multivariate linear regression. The mathematical explanation is clear and the results are complete. The language is One suggestion on the literature review is to add comparison with other covid forecast models. What makes this study unique when there exist other coronavirus projection models?

Experimental design

The methodology is clear especially in the explanation of velocity and acceleration related functions. The author could add more details, in both the paper and the attached code, on how the training set and test set are selected when applying cross-validation.

Validity of the findings

The covid cases and deaths are changing all the time, and it worth comparing and validating the forecast results with the current situation (from Sep 16, when the data was acquired, to the most recent dates). It will also be interesting to see how this model could be applied to other countries or even different Brazilian states.

Reviewer 2 ·

Basic reporting

the topic is good with the global recent interest.
using time series data is accepted at this level since the world still have not much information about covid -19. Furthermore, the article within the journal scope
However, the work has too many non necessary figures. and need more explanation and justification about the used model

Thus it is recommended to do major corrections the attached notes with the manuscript before accepting the article

Experimental design

NA

Validity of the findings

needs to validate the results

Additional comments

please respond the the comments attached with the original article

Annotated reviews are not available for download in order to protect the identity of reviewers who chose to remain anonymous.

Reviewer 3 ·

Basic reporting

Structure and overall presentation of the article can be improved a bit. A more comprehensive discussion regarding motivation and novelty of the study can improve its readability. Please explain a bit more how your contribution is different from the contribution of (Chen and Yu, 2020) .

Experimental design

The study is relevant to journal, however the novelty of the contribution seems not much high. I would like to see whether the proposed methodology will work efficiently using some other datasets (COVID cases from some other countries).

Validity of the findings

The results have been validated using the codes and data provided by the authors and they satisfactory. However, I am afraid that forecasts for next six months can be obtained with reasonable efficiency. in addition, the contents in the Figure 13 to Figure 16 are not clear. For example, there is no clarity about X.axis and Y-axis.

Additional comments

The manuscript is publishable after satisfactory response to the above comments.

Reviewer 4 ·

Basic reporting

Here are the comments:
• The language is sometimes difficult to read. The manuscript requires a complete rephrasing of sentences and correction of grammatical errors. The writing center on campus can be a useful resource.
• Equations 3 and 4 have a typo. The upper limit in the first integral and first summation should be (t+1). It is recommended to define F’(x) and H’(x) in equations 3 and 4 in a standard form instead of repeating the same formula. Similarly, the second derivative definitions in equations 5 and 6 can be written in a generalized form. Eqns 7 and 8 can be formatted in a single line.
• There are too many figures and tables. Figures 3 and 7-11 have x-axis and y-axis labels that are difficult to read.
• Few references of ARIMA, MA can be replaced by well-known books like Time Series Analysis: Forecasting and Control by Box and Jenkins.

Experimental design

• In the Methods section, standard procedures of time series analysis like exploratory analysis, decomposition, ACF, and PACF can be summarized more coherently and comprehensively. Adding a schematic diagram of the research process may be helpful.
• Data under the Results section can be relocated to the Methods section.
• Repetitive statements/conclusions can be avoided, e.g., The last line of the “Discussion” and “Conclusion” sections are the same.
• In the Results section, reporting error rates like Mean Absolute Error or root mean square error can be essential for choosing the best forecasting model and its accuracy accompanied by the already conducted exploratory analysis.

Validity of the findings

• Limitations of the study, future research directions, and comparison of results with background literature are suggested.

Additional comments

Overall, the manuscript is organized and represented in a well-structured manner. The results and figures were easily reproducible with the code and data provided.

---

## Round 0.2 · Minor Revisions

There are some minor comments to be addressed. Please revise the paper according to the reviewer's comments.

Reviewer 1 ·

Basic reporting

The revised manuscript provides more details in the background and literature review. It is good to see that this version updates the methods to support the conclusion.

I am confused by three versions of manuscripts in this revised package. The resubmitted articles in the Word document (peerj-53390-Tracked_Changes_PeerJ-research-manuscript.docx) and the PDF document (peerj-reviewing-53390-v1.pdf) are not exactly the same. The literature review sections are slightly different among these documents: the author’s letter (peerj-53390-Rebuttal_Letter.pdf), the Word document (peerj-53390-Tracked_Changes_PeerJ-research-manuscript.docx), and the PDF document (peerj-reviewing-53390-v1.pdf).

The following comments and line numbers are based on this document: peerj-reviewing-53390-v1.pdf.

Experimental design

1. About the cleaned function (Line 391). Is it a function removing possible outliers? If so, how you define the outliers? Visually clean out the outliers?
2. When comparing the weekly and monthly moving averages to the cleaned functions, the manuscript mentions that the moving averages are over-smoothing? How do you mathematically define over-smoothing?

Validity of the findings

1. About forecast. Instead of providing forecasts of the multivariate function and the composite function, could you provide forecasts of the real cases and deaths in Brazil and Indonesia? As the limitations section mentions, the current y-axis has little explanation for the forecast.
2. Instead of replicating the model in Indonesia, why not replicate the model in each Brazil state and see if the aggregation of the state-level forecast matches the country-level forecast? If we need to apply the same model for countries other than Brazil and Indonesia, why not apply it to more countries? I would see it is just a change of the input for each country. Without an explanation of randomly selecting Indonesia, I would doubt the external validity of this model.

Additional comments

1. Examples of format/grammar issues:
1) Capitalize “b” in “best” in this subtitle: Further Testing, Selecting the best Model, Forecasting and Analyzing (Line 449). Similar issue found in the Figure 7 caption.
2) Need a better sentence structure: The models are the Auto ARIMA Model without seasonality, a custom made ARIMA model that compensates for the serious lags found in the auto ARIMA model without seasonality, where necessary, named the adjusted lag model and the Auto ARIMA model with seasonality. (Lines 430-432)
3) Using the same indication for figures. For instance, when referring to a figure, the article sometimes uses Fig 2, while using Figure 2 in other places. (Lines 490-493)

2. Better graphs:
1) Use x-axis to label the exact dates in Figure 7 and 8 so the reader will know the real forecast time window. What do you mean by saying “dominant frequency” in the limitations section?
2) Correct the image distortion. Unify the labels, such as “a” and “(a).” Add axis labels in Figure 5, 7 and 8.

Reviewer 3 ·

Basic reporting

The manuscript has been improved and publishable now

Experimental design

The manuscript has been revised to my satisfaction

Validity of the findings

The manuscript has been revised to my satisfaction

Additional comments

I recommend the manuscript for publication

Reviewer 4 ·

Basic reporting

N/A

Experimental design

N/A

Validity of the findings

N/A

Additional comments

The authors have addressed all the comments.

---

## Round 0.3 · accepted · Accept

I agree that the paper can be accepted.

Reviewer 1 ·

Basic reporting

The basic reporting looks good. The author need to add the number in Line 50: On 30th June 2020, Brazil recorded - - the 2nd highest number of 50 confirmed cases of the coronavirus in the world.

Experimental design

Experimental design looks good.

Validity of the findings

Validity looks good and clear after the revision.